# Experimental Method for Evaluating the Reactivity of Alkali-Carbonate Reaction Activity

**DOI:** 10.3390/ma15082853

**Published:** 2022-04-13

**Authors:** Xiang Liu, Zhongyang Mao, Lei Yi, Zhiyuan Fan, Tao Zhang, Xiaojun Huang, Min Deng, Mingshu Tang

**Affiliations:** 1College of Materials Science and Engineering, Nanjing Tech University, Nanjing 211800, China; 201961203150@njtech.edu.cn (X.L.); mzy@njtech.edu.cn (Z.M.); 201961203130@njtech.edu.cn (L.Y.); 201961203172@njtech.edu.cn (Z.F.); 201961203162@njtech.edu.cn (T.Z.); 5967@njtech.edu.cn (X.H.); dengmin@njtech.edu.cn (M.D.); 2State Key Laboratory of Materials-Oriented Chemical Engineering, Nanjing Tech University, Nanjing 211800, China

**Keywords:** alkali-carbonate reaction, ACR activity, expansion, stress

## Abstract

The main aim of this study was focused on the Method of testing alkali-carbonate reaction activity to avoid alkali-carbonate reaction damage. In this paper, the alkali-carbonate reaction activity and alkali-silica reaction activity of ten kinds of aggregates were determined and analysed by existing standards and methods, by making specimens with aggregates of 2.5–5 mm and 5–10 mm particle size, cured in 1 mol/L tetramethyl ammonium hydroxide solution at 60 °C and 80 °C. Tetramethyl ammonium hydroxide solution was used to exclude the expansion caused by alkali-silica reaction. Effects of aggregate particle size and curing temperature on the expansion of samples were systematically investigated to determine alkali-carbonate reactivity of aggregates. In order to explore the relationship between stress and strain of aggregates, these aggregates were prepared into compacted bodies to test their stress and try to discover the pattern. The results showed that the expansion of the mould specimen prepared by the aggregate of 5–10 mm particle size, cured in 1 mol/L tetramethyl ammonium hydroxide solution at 80 °C was greater than 0.1% after 42 days, which could be used as a reference criterion to determine the alkali-carbonate reaction activity of the aggregate. In addition, the expansion stress test suggest that the alkali-carbonate reaction can generate expansion stress. The expansion stress of aggregates with alkali-carbonate reaction activity were much larger than that of aggregates without alkali-carbonate reaction activity. Through SEM and EDX analysis of the products of the alkali-carbonate reaction, it was shown that the dolomite crystals in the dolomitic aggregates reacted with the TMAH solution and resulted in alkali-carbonate reaction, forming calcite and brucite.

## 1. Introduction

Alkali-aggregate reaction [1,2,3] can be divided into alkali-carbonate reaction (ACR) and alkali-silica reaction (ASR). ACR also called alkali-dolomite reaction (ADR), which means that the alkali contained in the concrete pore solution reacts with the dolomite in the aggregate producing an abnormal expansion, making concrete crack and reducing the durability of the concrete [4,5,6]. Since Swenson [7,8] first discovered it in 1957, lots of research on the expansion mechanism of ACR has been carried out and developed methods to determine ACR activity of aggregates, mainly including petrographic method, rock prism method, concrete microbar method, concrete prism method and chemical method [9,10,11,12]. However, so far, no method was able to quickly and effectively detect aggregate ACR activity. The misjudgement of the alkali activity of aggregates also occurs in concrete engineering. Petrographic method describes the presence or absence of active ingredients in the aggregates. When the active ingredient is observed, other methods are needed to make further judgments. It is impossible to determine whether the aggregate has ACR activity, and it needs rich experience in petrography. For rock prism method, the long period, the obvious direction of rock prism cracking, the large difference of expansion value of rock prism, affected by ASR expansion and difficult to detect aggregate with slow expansion make it impossible to determine the ACR activity of carbonate aggregates accurately and effectively [13]. The problem of the concrete microbar method is that the influence of the siliceous component cannot be ruled out [14]. Namely, it cannot be determined whether the expansion is derived from ACR or ASR [15]. ASR expansion is not easy to produce due to low alkali concentration in concrete prism samples. However, the test period of concrete prism samples is very long, up to 360 days. It is difficult to determine the ACR activity of aggregates with slow expansion by chemical method. Therefore, it is necessary to develop a new method for determining the ACR activity of dolomitic aggregates.

Chen [16] reported alkali-silica reactive components such as microcrystalline quartz in sandstones and dolomitic rocks do not react with tetramethyl ammonium hydroxide (TMAH) solutions. Dolomite in the dolomitic rocks in TMAH solutions suffered from dedolomitization to form calcite and brucite. TMAH solutions may be used to distinguish ACR from ASR. According to Yang [17], samples with large particle size aggregates have larger expansion at high temperature, the rate of ACR can be significantly improved. Therefore, we can try to preserve the aggregates with alkali-carbonate reaction activity and the aggregates without alkali-carbonate reaction activity in TMAH solution, measure their expansion at different ages, and analyze and summarize the method to evaluate the alkali-carbonate reaction activity.

In this work, the ACR activity and ASR activity of ten kinds of aggregates were determined and analysed by RILEM AAR-2, RILEM AAR-5 and ASTM C1105 standard. Recording the expansion of mould specimen prepared by ten kinds of dolomitic aggregates of 2.5–5 mm and 5–10 mm particle size, cured in 1 mol/L TMAH solution at 60 °C and 80 °C at different ages. When the aggregate particle size is between 0.15 and 2.5, the particle size is too small, and the absolute volume of the expansion of a single aggregate is very limited, which limits the expansion of the specimen, and increases the particle size, the specimen can better reflect the ACR activity of the aggregate [18]. As the curing solution was TMAH solution and the curing temperature was higher, the test period of new method was short and the influence of silicon component was excluded, thus new method was not affected by ASR expansion. Powdered compacted body were prepared [19,20], their stress changes were tested, and the relationship between stress and strain of aggregates was analysed. Additionally, the cause of aggregate expansion stress was explored, and the reacted DH aggregate with dolomite enrichment area was selected for microscopic analysis.

## 2. Materials and Methods

### 2.1. Materials

The used materials were low-alkali Portland cement (Type II) obtained from the Jiangnan Cement plant Nanjing, China, with 0.54% equivalent Na_2_Oeq; Dolomitic limestones obtained from Baofuling Mountain, Weifang, Shandong, China (BFL), Duijieya Mountain, Sichuan, China (DJY), Zhencheng Mountain, Taiyuan, Shanxi, China (ZC), and Dolomitic rocks obtained from Wumi Mountain, Tianjin, China (WMS). Rocks SJW, CG, CX and SFP were derived from Guiyang, Guizhou, China. Rocks DH, JF were derived from Baoding, Hebei, China. The chemical compositions of the cement and dolomitic aggregates were tested according to GB/T 176-2017 (CIS, 2017), as shown in Table 1 and Table 2.

Figure 1 shows the XRD pattern of ten kinds of rocks, which are mainly composed of calcite (CaCO_3_), dolomite (CaMg(CO_3_)_2_) and quartz (SiO_2_). Figure 2 shows the distribution of dolomite crystals inside ZC and SFP rocks by polarizing microscopy. Figure 2a shows the distribution of dolomite in SFP rock is mosaic, many dolomite crystals are grouped together and embedded in the calcite matrix. Figure 2b shows the distribution of dolomite in ZC rock, where dolomite crystals are dispersed in the calcite matrix. The distribution of the two kinds of dolomite crystals is obviously different.

### 2.2. Methods

#### 2.2.1. Mortar Bars Test

According to RILEM AAR-2, rock aggregate was prepared in five grades, with particles of 0.16–4.75 mm, cement to aggregate ratio of 1:1.25, alkali content of cement adjusted to 0.90 wt% with NaOH, water cement ratio of 0.47, sample size of 25 mm × 25 mm × 280 mm. After the specimen was formed, it was placed in wet air at room temperature for curing, and demoulded 24 h later. The length of the specimen was measured and recorded as the initial length. Then, the specimen was cured in 1 mol/L NaOH solution at 80 °C. The specimen was taken out periodically and its length was measured after cooling to room temperature. RILEM AAR-2 stipulates that when the expansion of mortar specimen is less than 0.1% at 14 days, it is considered that the rock has no ASR activity. If the expansion is greater than 0.2% within 14 days, the aggregate is considered to have ASR activity. If the expansion is between 0.1% and 0.2%, it is considered to have potential ASR activity. When curing to a set age, remove the sample and measure the length, then calculate the expansion according to Equation (1):P_t_ = (L_t_ − L_0_)/(L_0_ − 2b) × 100%(1)
where P_t_ is the expansion after t days of curing, in %; L_t_ is the test piece length after t days of curing, in mm; L_0_ is the initial test piece length, in mm; and b is the length of the nail embedded in the concrete, in mm.

#### 2.2.2. Concrete Microbars Test

Concrete microbars prepared according to RILEM AAR-5 were 40 × 40 × 160 mm in size with aggregate of 5–10 mm particle size and aggregate to cement ratio of 1:1. After the specimen was formed, it was placed in wet air at room temperature for curing, and demoulded 24 h later. The length of the specimen was measured and recorded as the initial length. Then the concrete microbars were cured in 1 mol/L NaOH solution at 80 °C, and the specimens were taken out periodically. After cooling to room temperature, the length was measured. RILEM AAR-5 stipulates that if the expansion of concrete microcolumns is less than 0.1% within 28 days, the rock is considered to have no ACR activity. If the expansion is more than 0.1% within 28 days, the aggregate is considered to have ACR activity. The equation of expansion is shown in Equation (1).

#### 2.2.3. Concrete Prisms Test

Tests were conducted according to ASTM C1105. The alkali equivalent (equivalent Na_2_Oeq) of the cement was adjusted to 1.8 kg/m^3^. The size of concrete prisms was 75 mm × 75 mm × 285 mm, and each sample was formed into three specimens. After moulding, the specimen was cured in an environment of 20 ± 0.5 °C and relative humidity ≥95%, the mould was removed for 24 h, and placed in a curing box of 38 ± 0.5 °C and RH = 100%. One day later, the initial length of the specimen was measured. The specimens were taken out at 7, 28, 56, 90, 180, 270 and 360 days and cooled to 20 °C for length measurement. The equation of expansion is shown in Equation (1).

#### 2.2.4. New Experimental Method for Evaluating Alkali-Carbonate Reaction Activity

The aggregate is broken into 2.5–5 mm and 5–10 mm particle size, and put 650 g of them into a leaky bucket mould with a diameter of 9 cm, a height of 11 cm and a inner hole diameter of 1.2 mm. Adjust the parameters of the block press made in Nanjing, China (6 MPa, the holding time is 5 s), and compact the rock particles. A small amount of cement is used to fix the nail head vertically in the middle of the upper rock grain. All specimens were kept for 24 h in a moisturized condition (RH = 98%) for curing before demoulding. Curing conditions for the samples made with the aggregates with ACR activity were: *t* = 60 and 80 °C, and c = 1 N (concentration of TMAH). The equation of expansion is shown in Equation (1).

#### 2.2.5. Expansion Stress Test

In order to test the expansion stress of the aggregate producing the alkali-carbonate reaction, a stress test apparatus was designed and assembled as shown in Figure 3. The aggregate of 5–10 mm particle size was used in the experiment. Then 40 g of aggregate was put into the mould and the aggregate was compacted with a press. The pressure value was 40 MPa and the holding time was 5 s. We then assembled the mould, the pressure was adjusted to 15 ± 0.1 MPa by tightening the nut, cured in 1 mol/L TMAH solution at 80 °C, recording the expansion stress. The equation of expansion stress is shown in Equation (2).
(2)$$\sigma =\frac{4\left(F_{t}-F_{0}\right)g}{\pi d^{2}}$$
where *σ* is the expansion stress (MPa); *F_t_* is the sensor value at time t (kg); *F*_0_ is the initial value of the sensor (kg); *g* is the gravity acceleration, which has a value 9.8 m/s^2^; *d* is the inner diameter of the mould, which has a value of 24 mm; and *π* is 3.14.

#### 2.2.6. Analysis by SEM, Element Mapping and X-ray Diffraction

In this paper, SEM and element mapping were used to analyze the morphology of ACR products of dolomitic rocks, and X-ray diffraction was used to analyze aggregate reaction products. The aggregate in the stress test apparatus was preserved in 80 °C and 1 mol/L TMAH solution. After sample preparation, small pieces of dolomite rich area on the rock fracture surface were sprayed with gold, and then the product morphology was observed under the scanning electron microscope made in Tokyo, Japan. Among them, the acceleration voltage of scanning electron microscope is 20 kV, and the acceleration current is 5 mA. Aggregate is grounded into powder and the XRD data are collected in the range of 5–80°, 2θ at a counting time of 15 s/step and a divergence slit of 1°. The XRD measurement instrument is SMART LAB made in Tokyo, Japan.

## 3. Results and Discussion

### 3.1. Discrimination of the Aggregate Alkali Reaction

The results of the mortar bar test and concrete microbar test are shown in Figure 4 and Figure 5, respectively. Figure 4 shows that the expansion of DH, DJY, SFP, WMS and ZC at 14 days was 0.016%, 0.010%, 0.022%, 0.022% and 0.017%, respectively. Expansion here refers to volume expansion. The dotted line in Figure 4 is the threshold value for judging whether the aggregate demonstrates an alkali-silica reaction. At 14 d, the expansion of DH, DJY, SFP, WMS and ZC were less than 0.1%; therefore, according to RILEM AAR-2, the DH, DJY, SFP, WMS and ZC aggregates had no alkali-silica reaction activity. Mortar specimens prepared from rock BFL, CG, CX, JF and SJW have expansion of 0.14%, 0.17%, 0.17%, 0.12% and 0.11% at 14 d, respectively, greater than the threshold value of 0.1% but less than 0.2%. Therefore, rock BFL, CG, CX, JF and SJW were judged to have potential ASR activity.

Figure 5 shows the expansion of concrete microbars cured in 1 mol/L NaOH solution at 80 °C. As shown in the figure, the expansion of the samples cured in a NaOH solution increased significantly at different ages. The expansion of the samples prepared by BFL, CG, CX, JF, SJW, SFP, WMS and ZC at age 28 d were 0.17%, 0.25%, 0.27%, 0.18%, 0.18%, 0.12%, 0.12%, 0.11%, respectively. According to the RILEMAAR-5 standards, BFL, CG, CX, JF, SJW, SFP, WMS and ZC were affected by ACR activity. Especially after 28 days, the expansion of the specimen increases rapidly. At the later stage, the expansion of the concrete become stable. We can see that the expansion of the concrete microbars prepared by DH and DJY were less than 0.1% at age 28 d. According to RILEM AAR-5, the DH and DJY aggregates had no ACR activity. The expansion of concrete microbars prepared from BFL, CG, CX, DH, DJY, JF, SJW, SFP, WMS and ZC at 84 d were 0.35%, 0.52%, 0.44%, 0.25%, 0.071%, 0.43%, 0.33%, 0.29%, 0.27% and 0.27%, respectively.

### 3.2. Discrimination of Concrete Prisms Test

The problem of concrete microbars test is that it cannot exclude the influence of siliceous composition and cannot determine whether the expansion of aggregate comes from alkali-carbonate reaction or alkali-silica reaction. Therefore, there is a certain deviation when determine the ACR activity of aggregates by concrete microbars test. In order to further determine the ACR activity of various rock aggregates, the expansion law of concrete prisms prepared by different aggregates according to ASTM C1105 standard was studied, and the alkali activity of aggregates was determined, so as to compare with the determination results of concrete microbars method.

Figure 6 shows the expansion curve of concrete prisms prepared with dolomitic aggregate. The expansion of the concrete prisms prepared by CG, CX, JF, SFP, WMS and ZC at age 360 d was 0.41%, 0.81%, 0.12%, 0.076%, 0.045% and 0.051%, respectively, greater than the threshold value of 0.03%. On the contrary, the BFL, DH, DJY and SJW samples expanded about 0.029%, 0.029%, 0.023% and 0.027%, respectively, at 360 days, were less than 0.03%, which is indicated by the position of the dotted line in the Figure 6. According to the ASTM C1105 standards (the expansion of the specimen is greater than 0.03% at 360 days), the BFL, DH, DJY and SJW aggregates had no ACR activity, while the aggregates of CG, CX, JF, SFP, WMS and ZC demonstrated ACR activity. According to the RILEMAAR-5 standards, BFL, CG, CX, JF, SJW, SFP, WMS and ZC were affected by ACR activity, the result was different from discrimination of concrete prisms test. It should be noted that the concrete microbars test cannot exclude expansion caused by alkali-silica reaction and cannot effectively identify the ACR activity of aggregates. The ASTM C1105 standard states that alkali-silica reaction does not occur at low alkali concentration, so it can be used to accurately determine the ACR activity of aggregates.

### 3.3. Discrimination of New Experimental Method for Evaluating Alkali-Carbonate Reaction Activity

The test period of concrete prism method is as long as one year. Although the result is accurate, it affects the project construction. The new method uses 1 N TMAH and 20 g of cement to minimize the alkali-silica reaction of the aggregate, so that the aggregate expansion comes entirely from the alkali-carbonate reaction.

Figure 7 shows the expansion of ten kinds of rocks with 2.5–5, 5–10 mm particle size at 60 °C in 1 mol/L TMAH solution. From Figure 7, the expansion process of aggregates can be divided into three stages. For the first stage, the aggregates appear fast expansion. The expansion of samples at 14 days were more than 0.05%. Compared with samples made by 2.5–5 mm aggregates, 5–10 mm aggregates had a higher expansion at early curing ages. The second stage, samples had obvious expansion as curing ages increase and the differences in expansion began to expand. Final stage, the expansion of most of aggregates appeared slow and tended to be stable. At 42 days, the expansion of BFL, DH, DJY and SJW with 2.5–5 mm particle size at 60 °C were 0.082%, 0.064%, 0.083% and 0.096%, less than 0.1%, but the expansion of DJY and SJW were up to 0.11% and 0.12%, more than 0.1% at 70 days. From Figure 7b, the expansion process of examples was similar to that in Figure 7a. Compared with the samples made by 2.5–5 mm aggregates, 5–10 mm samples had bigger expansion, and the expansion of all of samples with 5–10 mm particle size were more than 0.1% at 70 days.

In addition to the particle size of aggregates, curing temperature also affects the expansion of samples. Figure 8 shows the expansion of samples prepared by aggregates of 2.5–5 mm and 5–10 mm particle size, cured in 1 mol/L TMAH solution at 80 °C. The expansion data of specimens at 60 °C and 80 °C were analysed, the expansion of most of examples cured at 80 °C was about 30% larger than that cured at 60 °C. From Figure 8, compared with 2.5–5 mm samples, the differences in the expansion of 5–10 mm samples were more obvious at early curing age. When the curing age was 42 days, the expansions of BFL, DH, DJY and SJW with 2.5–5 mm particle size cured at 80 °C were 0.087%, 0.097%, 0.099% and 0.075%, respectively, were less than 0.1%, and tended to be stable. The expansions of them were greater than 0.1% cured at 84 days curing age.

According to the ASTM C1105 standards, the BFL, DH, DJY and SJW aggregates had no ACR activity. From Figure 8b, the expansion of BFL, DH, DJY and SJW with 5–10 mm particle size cured at 80°C were 0.098%, 0.098%, 0.086% and 0.12% at 56 days, and their expansion were all less than 0.1% at 42 days. The expansion of samples prepared by all aggregates with ACR activity were more than 0.1% at 42 days. It can be seen from Figure 7 and Figure 8, when the curing temperature and particle size increased, the expansion of samples became larger. The difference of expansion between aggregates with and without ACR activity increases with the increase of temperature and particle size. Therefore, the conclusion can be drawn: the expansion of the specimen is greater than 0.1% at 42 days in 1 mol/L TMAH solution under curing condition of 80 °C, which can be used as a reference criterion to determine the alkali-carbonate reaction activity.

### 3.4. Expansion Stress

Figure 9 shows the expansion stress curve of ten kinds of aggregates with 5–10 mm particle size compacted body cured in 1 mol/L TMAH solution. The expansion stress continuously increased slowly in front of the 28 days, and increased gradually after a brief plateau. After 42 days, the expansion stress growth rate increased rapidly, and the 105 days expansion stress of CG, JF, CX, ZC, SFP and WMS samples prepared by these aggregates with ACR activity had reached 145, 116, 112, 111, 78 and 71 MPa, respectively. As seen from Figure 9, there was huge difference in the development of the expansion stress of the compacted bodies between aggregates had no ACR activity and aggregates had ACR activity. In the TMAH solution, the development of expansion stress was roughly divided into three stages, namely, the slow growth period, the stationary period and the rapid growth period. During the slow growth period, the agglomeration of dolomite particles led to the slow growth of the dolomite compacted body, which led to a small increase in the expansive stress. During stabilization, the dolomite crystals react with TMAH solution, causing the dolomite compacted body to expand, and the expansion of the compacted body just counteract its volume contraction. With the increase of curing time, the degree of chemical reaction intensified and the expansion stress of the dolomite compacted body increased gradually. After 28 days, the growth rate increased faster and faster, but the expansion stress of BFL, DH, SJW and DJY samples prepared by these aggregates with no ACR activity reached 48, 42, 35 and 24 MPa, respectively, on 105 days in the final.

### 3.5. Products Analysis

From Katayama [21,22], according to polarizing microscopy, it was observed that ACR produced a myrmekitic texture, which was composed of spotted brucite and calcite within the reaction rim. As shown in Figure 10, in order to further study the cause of expansion stress caused by alkali-carbonate reaction, DH aggregate with dolomite rich areas and cracks after reaction was selected for observation through SEM and element mapping analysis. As can be seen from Figure 10a, there are many rod-like crystals next to the calcite. Figure 10b–e shows the element mapping analysis identified in Figure 10a. The combination of these two images indicates that rod-like brucite crystals do exist in ACR process. The reaction products of dolomite with alkali in aggregate were brucite and calcite. It is worth noting that unreacted dolomite is still present in aggregate particles because some of the dolomite has not been fully reacted in the dedolomitization process. ACR can continue to occur when the pH and temperature of the curing solution are high enough. Since the TMAH solution used in this study excludes the effect of microcrystalline quartz and excludes the expansion contribution of ASR, the expansion only comes from ACR. When the region is mainly composed of Ca, C and O elements, this region is calcite (CaCO_3_) produced by ACR. In addition, when the region is mainly composed of Mg and O elements, this region is brucite (Mg(OH)_2_) produced by ACR. The stacking and growth of brucite crystal produce expansion stress, and the aggregate undergo alkali-carbonate reaction to produce expansion.

## 4. Discussion

The new method studied in this paper is quite different from rock prism method, concrete microbar method and concrete prism method for evaluating ACR activity. First of all, the new method uses TMAH as curing solution, which excludes the influence of siliceous component, so that the expansion of aggregates excludes ASR expansion, thus the expansions observed are all due to ACR expansion. Both the new method and the concrete prism method can determine that the BFL, DH, DJY and SJW aggregates had no ACR activity, while the concrete microbar method can only determine that the DH and DJY aggregates had no ACR activity under the influence of ASR expansion. Secondly, the disadvantage of the rock prism method is that the uneven distribution of dolomite will lead to a large difference in the expansion of these rock prisms. Therefore, the raw material of the new method is aggregate of 5–10 mm particle size, so the new method will not have such disadvantage. In addition, the curing temperature of the new method is 80 °C, which increases the expansion of the specimen, so that the test period of the new method is reduced to 42 days, which is smaller than the test period of rock column method (90 days) and concrete prism method (360 days). Of course, the new method has many shortcomings. In this study, ten kinds of aggregates were used to explore a new method to evaluate ACR activity, and more aggregates are needed to verify the accuracy of the new method. In addition, the new method needs to be compared with the existing methods to verify its reliability. The experimental period of the new method is longer than that of the 28-day concrete microbar method. The experimental period of the new method can be reduced by changing the curing solution concentration, the curing solid-liquid ratio and the curing temperature.

## 5. Conclusions

In this paper, the ACR activity and ASR activity of various aggregates were determined and analysed by existing standards and methods. By reducing the amount of cement and curing the aggregates in 1 mol/L TMAH solution, thus expansion of aggregate is attributed to alkali-carbonate reaction. The influence of curing temperature and aggregate size on the expansion of dolomitic aggregate was systematically studied by using self-made mould specimen prepared from ten kinds of dolomitic aggregates. The following main conclusions can be drawn from physical measurements and microstructure analysis.

Firstly, the ACR rate can be significantly increased by changing the particle size of the aggregates and increasing the curing temperature, thus accelerating the expansion of the sample. Compared with 2.5–5 and 5–10 mm particle size, most of samples prepared with 5–10 mm particle size have bigger expansion. The expansion of samples cured at 80 °C was 30% larger than that cured at 60 °C for 84 days. The expansion of the mould specimen prepared by the aggregate of 5–10 mm particle size, cured in 1 mol/L TMAH solution at 80 °C was greater than 0.1% after 42 days, which could be used as a reference criterion to determine the ACR activity of the aggregate.

Based on the analysis of expansion stress curve of compacted bodies, the expansion stress of aggregates with ACR activity were much larger than that of aggregates without ACR activity. The maximum expansion stress was as high as 145 MPa and the minimum is as low as 24 MPa on 105 days. The SEM and element mapping analysis indicated that rod-like brucite crystals were formed in the process of ACR. The reaction products of ACR, including brucite and calcite, were distributed around dolomite crystals.

## Figures and Tables

**Figure 1 materials-15-02853-f001:**
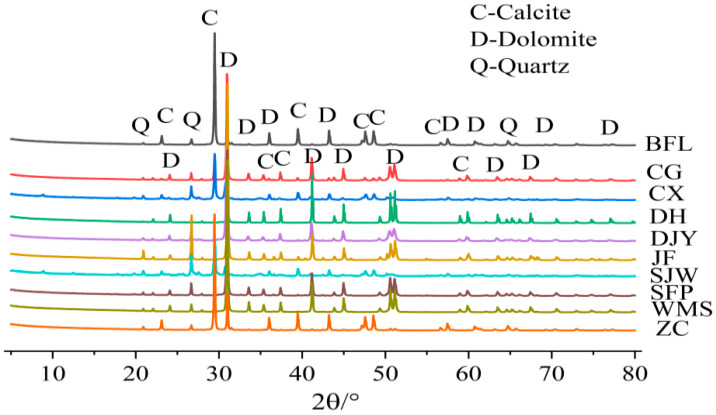
The XRD pattern of different rocks.

**Figure 2 materials-15-02853-f002:**
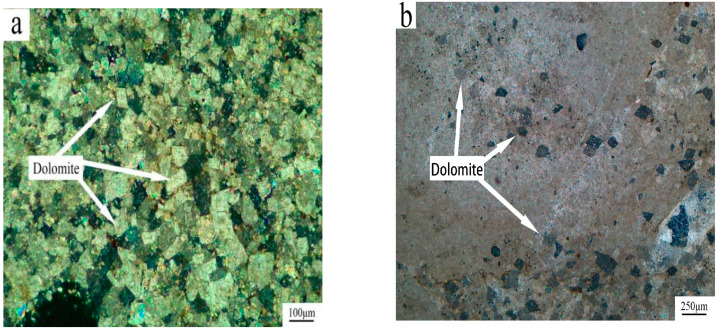
The distribution of dolomite crystals inside SFP and ZC rocks: (**a**) SFP mosaic distribution, (**b**) ZC dispersive distribution.

**Figure 3 materials-15-02853-f003:**
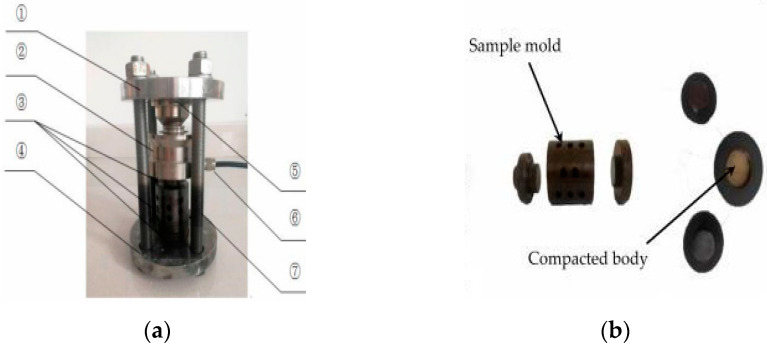
A schematic diagram of the expansion stress testing apparatus (①: top plate; ②: sensor; ③: sample mould; ④: bottom plate; ⑤: anti-load measuring head; ⑥: data acquisition system; ⑦: constrained screw). (**a**) Expansion stress test apparatus; (**b**) Sample mould.

**Figure 4 materials-15-02853-f004:**
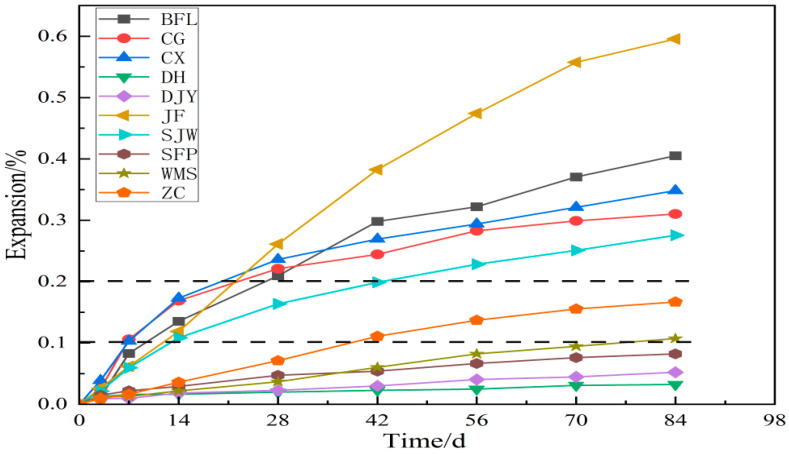
The expansion of the mortar bars prepared according to RILEM AAR-2.

**Figure 5 materials-15-02853-f005:**
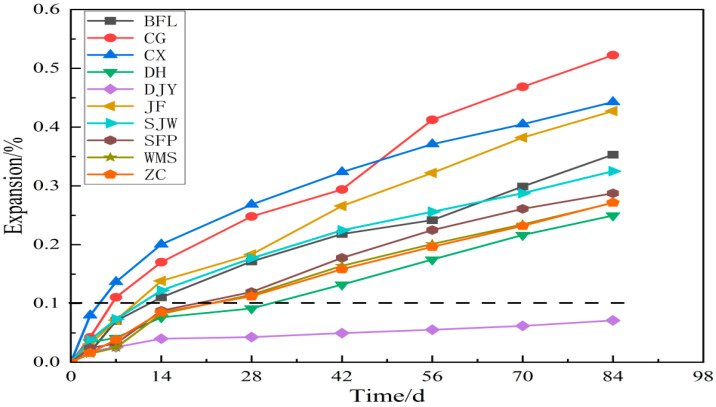
The expansion of the concrete microbars prepared according to RILEM AAR-5.

**Figure 6 materials-15-02853-f006:**
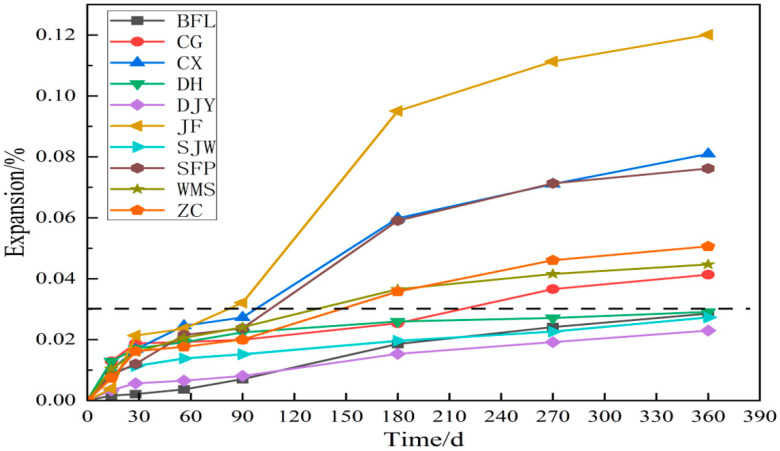
The expansion of the concrete prisms prepared according to ASTM C1105.

**Figure 7 materials-15-02853-f007:**
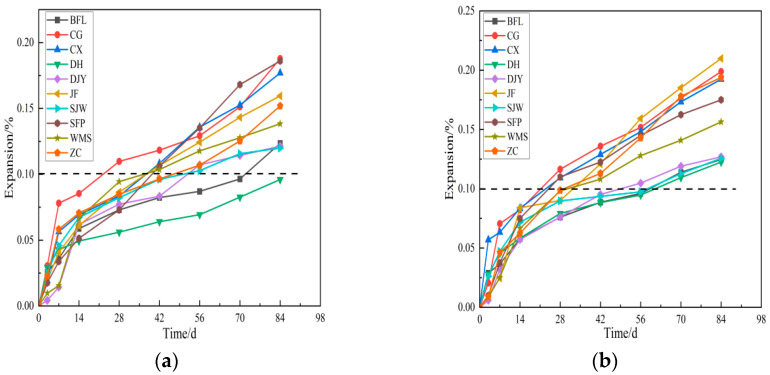
The expansion of rocks with (**a**) 2.5–5 mm, (**b**) 5–10 mm particle size at 60 °C.

**Figure 8 materials-15-02853-f008:**
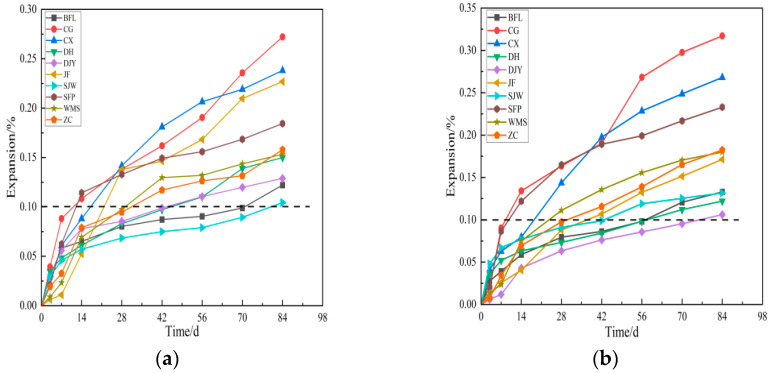
The expansion of rocks with (**a**) 2.5–5 mm, (**b**) 5–10 mm particle size at 80 °C.

**Figure 9 materials-15-02853-f009:**
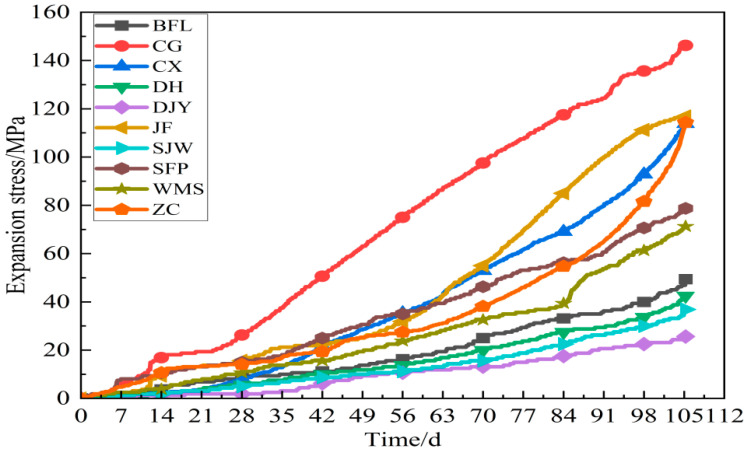
Expansion stress curve of compacted bodies.

**Figure 10 materials-15-02853-f010:**
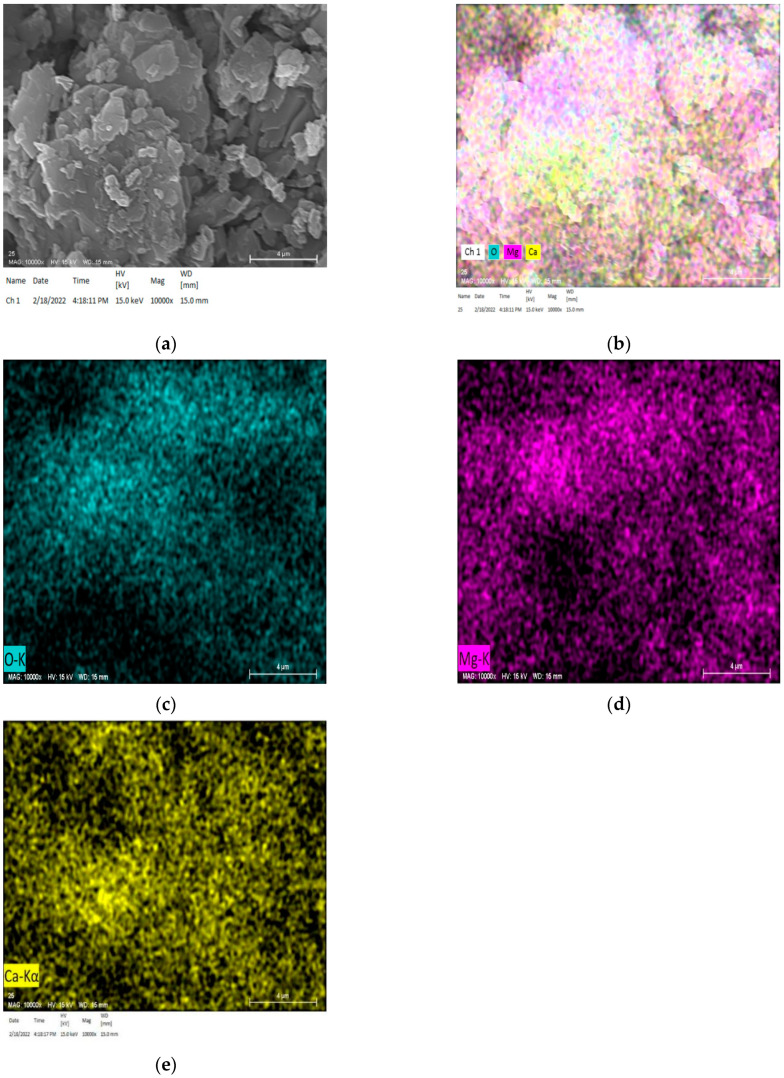
SEM, mapping element analysis. (**a**) SEM image of DH aggregate with 5–10 mm grain cured in TMAH at 80 °C for 105 days. (**b**) mapping analysis determined in Figure 10a. (**c**) O images. (**d**) Mg images. (**e**) Ca images.

**Table 1 materials-15-02853-t001:** Chemical analysis of cement (wt.%).

Samples	Chemical Analysis of Cement (wt.%)								
	SiO_2_	CaO	MgO	Al_2_O_3_	Fe_2_O_3_	SO_3_	K_2_O	Na_2_O	LOI
Cement	22.02	60.51	2.18	6.34	3.05	1.86	0.47	0.23	1.96

**Table 2 materials-15-02853-t002:** Chemical analysis of aggregates (wt.%).

Samples		Chemical Compositions (wt.%)					
		SiO_2_	CaO	MgO	Al_2_O_3_	Fe_2_O_3_	LOI
1	BFL	4.00	33.65	16.10	0.80	0.70	1.45
2	CG	20.15	33.92	4.85	4.97	2.86	2.14
3	CX	18.17	32.05	4.97	5.79	2.85	4.51
4	DH	0.63	29.13	21.85	0.24	0.33	1.63
5	DJY	6.78	29.83	16.47	1.04	1.49	1.12
6	JF	10.51	26.50	19.10	0.24	0.33	2.01
7	SJW	26.14	25.85	5.10	8.60	3.63	4.52
8	SFP	3.83	28.38	20.53	0.25	0.50	2.03
9	WMS	0.18	29.99	22.61	0.12	0.32	0.16
10	ZC	2.38	47.03	5.13	0.24	0.33	1.42

## Data Availability

The data presented in this study are available on request from the corresponding author.

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
