# Peer review of "Experimental Method for Evaluating the Reactivity of Alkali-Carbonate Reaction Activity"

_materials, 2022, doi:10.3390/ma15082853_

Round 1

Reviewer 1 Report

materials-1667814

Article title:

Experimental method for evaluating the reactivity of alkali-carbonate reaction activity

Comments:

Great efforts have been performed with sufficient testing and nice outcomes, but some issues and typos need to be clarified.

  1. The main aim of this work was focused on the “Method of testing ACR”. Please the authors include this point in the abstract section.
  2. In the introduction section, a short summary of the differences or benefits of the existing methods and the new method should be addressed.
  3. Please include the technique of chemical composition analysis, Tab 1 and 2.
  4. Line 86, the XRD results, the full chemical formula of C calcite, D dolomite, and Q quartz should be presented.
  5. Figure 2, Fig alignment.
  6. Figures 7 and 8, the vertical number of a & b should be on the same scale e.g., 0.35 for all, to easily compare the results.
  7. Figure 9, the symbol markings are required, just in case, for the black and white printing.
  8. Section 3.5 and the conclusion, the discussion, and concise explanation on what are the differences, benefits, drawbacks, etc. of the new purpose method are strongly required rather than expansions, stress, and SEM images.
  9. If possible, the XRD results of hardened products should be present and discussed.

Reviewer 2 Report

Please find attached my comments
